# Recombinant Klotho Protein Ameliorates Myocardial Ischemia/Reperfusion Injury by Attenuating Sterile Inflammation

**DOI:** 10.3390/biomedicines10040894

**Published:** 2022-04-13

**Authors:** Jinwoo Myung, Jin-Ho Beom, Ju-Hee Kim, Ji-Sun Woo, Incheol Park, Sung-Phil Chung, Yong-Eun Chung, Je-Sung You

**Affiliations:** 1Department of Emergency Medicine, Yonsei University College of Medicine, Seoul 06273, Korea; jwm1125@yuhs.ac (J.M.); wangtiger@yuhs.ac (J.-H.B.); juheekim55@yuhs.ac (J.-H.K.); jisunwoo224@yuhs.ac (J.-S.W.); incheol@yuhs.ac (I.P.); emstar@yuhs.ac (S.-P.C.); 2Department of Radiology, Yonsei University College of Medicine, Seoul 06273, Korea

**Keywords:** acute myocardial infarction, klotho, myocardial ischemia/reperfusion injury, high mobility group box-1, sterile inflammation

## Abstract

Currently, no effective therapy and potential target have been elucidated for preventing myocardial ischemia and reperfusion injury (I/R). We hypothesized that the administration of recombinant klotho (rKL) protein could attenuate the sterile inflammation in peri-infarct regions by inhibiting the extracellular release of high mobility group box-1 (HMGB1). This hypothesis was examined using a rat coronary artery ligation model. Rats were divided into sham, sham+ rKL, I/R, and I/R+ rKL groups (*n* = 5/group). Administration of rKL protein reduced infarct volume and attenuated extracellular release of HMGB1 from peri-infarct tissue after myocardial I/R injury. The administration of rKL protein inhibited the expression of pro-inflammatory cytokines in the peri-infarct regions and significantly attenuated apoptosis and production of intracellular reactive oxygen species by myocardial I/R injury. Klotho treatment significantly reduced the increase in the levels of circulating HMGB1 in blood at 4 h after myocardial ischemia. rKL regulated the levels of inflammation-related proteins. This is the first study to suggest that exogenous administration of rKL exerts myocardial protection effects after I/R injury and provides new mechanistic insights into rKL that can provide the theoretical basis for clinical application of new adjunctive modality for critical care of acute myocardial infarction.

## 1. Introduction

Acute myocardial infarction (AMI) is a common cardiac emergency associated with high rates of morbidity and mortality worldwide [1]. AMI is characterized by myocardial cell death via time-dependent ischemia by prolonged occlusion of the infarct-related artery [2,3]. To minimize necrosis of the myocardium mediated by the infarct-related artery (IRA), the area at risk (AAR) and blood flow to the IRA should be restored as soon as possible via mechanical reperfusion with thrombolytic therapy and a coronary artery stent [2,3,4]. For safe and effective treatment, current guidelines recommend primary percutaneous coronary intervention (PCI) as the preferred reperfusion strategy that should be performed for patients with ST elevation myocardial infarction (STEMI) within 12 h of symptom onset, which is considered as the therapeutic window [3,5]. In developed countries, the incidence of STEMI has reduced over the last two decades [6]. The door-to-balloon time has remarkably improved with advances in medical science [1,6]. Despite widespread reperfusion therapy, mortality in patients with STEMI is substantial, as is the development of heart failure among the older population [6]. Although the incidence of STEMI is increasing in developing countries, interventional strategies for reperfusion are often not available [6]. In addition, timely success of reperfusion is the most effective strategy in limiting infarct size and subsequent ventricular remodeling [6]. Reperfusion comprises an additional component of irreversible myocardial injury and disturbance of coronary circulation and, subsequently, contributes to the final infarct size [6]. New treatments are needed to reduce myocardial infarct size and to preserve left ventricular function. Studies examining treatment for myocardial reperfusion injury have not focused on finding a subsidiary therapeutic target. Myocardial reperfusion injury refers to cardiomyocyte death that paradoxically results from reperfusion of ischemic myocardium [7]. Currently, no effective therapy is available, and a potential target for preventing myocardial ischemia and reperfusion injury in patients with STEMI has not been identified [8,9].

Sterile inflammation has recently emerged as an important aggravator in the early stage of ischemic/reperfusion (I/R) injury. Therefore, new adjunctive therapies are required against AMI to widen the therapeutic window and ameliorate early myocardial damage induced by I/R injury. Myocardial I/R injury is very complex, and the pathophysiology is also poorly understood. In myocardial damage induced by I/R injury, inflammatory response induction and cell death by apoptosis play critical roles in the propagation of ischemic injury [10,11,12,13]. High mobility group box-1 (HMGB1) serves as the structural organizer of DNA in eukaryotic cells. Upon I/R injury, this protein is rapidly released, and its levels are increased after 30 min of ischemia [14]. HMGB1, which is one of the primary mediators of the innate immune response, is passively released by necrotic cells or actively released during I/R injury [15]. Binding of extracellular HMGB1 to toll-like receptor 4 (TLR-4) activates several inflammatory mediators that amplify and expand the extent of damage after I/R and induce cardiomyocyte apoptosis after I/R [15] Administration of glycyrrhizin, a known pharmacological inhibitor, which binds directly to HMGB1 and blocks extracellular release of HMGB1, significantly decreases the infarct size and the levels of serum HMGB1 as well as inflammatory cytokines including interleukin (IL)-1β, IL6, and tumor necrosis factor (TNF)-α [10,16]. Clinically, increased levels of HMGB1 in plasma were significantly associated with a higher risk of mortality in patients with STEMI receiving PCI [17]. This insight into the pathophysiological features of acute STEMI expands the therapeutic scope of this disease beyond the traditional strategies that focus on reducing stenosis.

Klotho, which is a membrane-bound, soluble, and anti-aging protein, demonstrates protective activities in multiple organs [18]. In klotho-deficient mice, inherited phenotypes resemble aging in humans, whereas the lifespan is extended in animals with overexpression of klotho [19,20]. Klotho is a key factor in fibrous growth factors’ (FGF) signaling and is associated with the modulation of mineral metabolism and energy [21,22]. Exogenous kl gene insertion into cells leads to significant attenuation of apoptosis [23]. Moreover, this protein acts as a hormone exerting anti-inflammation and antioxidative effects, thereby regulating inflammation, oxidative stress, and fibrosis via attenuation of transforming growth factor-β1 (TGF-β1) and insulin/insulin-like growth factor-1 (IGF-1) signaling pathways [22,24,25]. In muscle progenitor cells, decreased expression of α-Klotho is associated with increased cytoplasmic expression of HMGB1 [26]. It is critical to understand the direct functional and mechanistic links between klotho and HMGB1 in case of inflammation after I/R injury in a clinically relevant model of AMI.

Several recent studies have demonstrated that klotho deficiency is related to the development of cardiomyopathy and cardiovascular diseases, and a higher concentration of plasma klotho was an independent predictor of a lower likelihood of cardiovascular disease [27,28,29]. Until now, although some scientists have focused on the role of klotho in the development of cardiovascular diseases, their studies were limited to the pathophysiology of chronic cardiovascular diseases. No effective study has been published examining the therapeutic effectiveness of direct administration of recombinant klotho (rKL) in early myocardial damages after I/R injury. We hypothesized that rKL protein administration could attenuate sterile inflammation in peri-infarcted regions by inhibiting the extracellular release of HMGB1 and aimed to examine this hypothesis using the left anterior descending coronary artery (LAD) ligation rat model. Our findings demonstrated subsequent reduction in the myocardial infarct area after I/R injury. The present study investigated whether the administration of rKL protein exerts a myocardial protective effect after I/R injury with the same underlying mechanism.

## 2. Materials and Methods

### 2.1. Preparation of Experimental Animals

Animal care and all experiments of this study were performed in strict accordance with protocols and guidelines approved by the National Institutes of Health and the Institutional Animal Care and Use Committee of the Yonsei University Health System (2018-0262). Nine-week-old Wistar rats were provided by a company (Orientbio, Seongnam, Korea), and the Wistar rats were used after an acclimatization period of approximately 1 week, according to the standards of this research institute. We used 102 healthy, male, age-matched Wistar rats weighing 400–430 g in this study (Appendix A). All the animals were acclimatized in plastic cages under sterile conditions, with a 12-h/12-h light/dark cycle, 50 ± 10% humidity, and a temperature of 22 ± 2 °C, for 1 week. During this period, the rats were fed a standard diet, with fresh water supplied ad libitum [10].

### 2.2. Experimental Rat Model of Myocardial I/R Injury

We anesthetized the rats with 5% isoflurane, which was administered as a gas mixture of oxygen (0.3 L/min) and nitrous oxide (0.7 L/min). During the surgery, we also maintained the anesthetic status of the rats with 2% isoflurane administered in the same gas mixture. After a midline neck incision, tracheostomy of the rats was conducted using an intravenous catheter (4712-020-116. I.V Catheter 16 G, Sewoon Medical Co., Cheon-An, Korea) as the tracheostomy tube. A rodent ventilator (SAP-830/AP, CWE, Inc., Ardmore, PA, USA) supported mechanical ventilation (tidal volume, 3.0 mL; respiratory rate, 50/min) during the surgery. After left vertical thoracotomy, we exposed the heart via pericardiectomy. As described in a previous study, we ligated the LAD coronary artery at the mid-portion between the pulmonary artery and apex in the exposed heart with a 6-0 ethilon suture [30]. Immediately before ligation, we inserted the PE-10 tube (polyethylene tube, OD 0.61 mm) between the suture and LAD and ligated the suture with the PE-10 tube. We visually confirmed that ischemia in the form of cyanosis and dyskinesia of the myocardium induced by LAD had developed after ligation. After 30 min of LAD ligation, we induced reperfusion by removing the PE-10 tube and stitched the skin with 4-0 nylon sutures after reperfusion. We also performed the same surgical procedures, except for ligation, in sham animals [30]. After 4 or 24 h of LAD ligation, we administered anesthesia via inhalation of 5% isoflurane in a mixture of 0.7 L/min nitrous oxide and 0.3 L/min oxygen and euthanized the animals [10].

### 2.3. Experiment Protocol

To assess the optimal dose of rKL for myocardial protection, first, the rats were randomly divided into four experimental groups: sham, MI + 0.025 µg of rKL/g of body mass, MI + 0.05 µg of rKL/g of body mass, and MI + 0.075 µg of rKL/g of body mass. We identified the infarct volume by 2,3,5-triphenyltetrazolium chloride (TTC) staining in the four groups. Thus, we determined that 0.05 µg/g was the optimal dose of rKL for myocardial protection. We subsequently used this optimal dose of rKL for myocardial protection. To identify the protective effects of klotho administration against myocardium after myocardial I/R injury, the rats were randomly divided into four experimental groups: sham, sham + klotho, MI, and MI + klotho. A total of 0.05 µg/g of rKL in 0.5 mL of saline was also administered via intraperitoneal injection 15 min after LAD ligation. The same volume of saline was administered via intraperitoneal injection to control groups. Reperfusion was induced after 15 min of rKL administration. During all experiments with rats, we closely monitored their core temperature and maintained it at 37.0 ± 0.5 °C, via their rectum, with a controlled heating pad with a feedback function (HB 101, Harvard Apparatus, Holliston, MA, USA).

### 2.4. Assessment of Infarct Volume

To identify myocardial infarction, we stained the infarcted area with 2,3,5-triphenyltetrazolium chloride (TTC) (T8877, Sigma-Aldrich, St. Louis, MO, USA) at 4 or 24 h after LAD I/R. We re-opened the chest of anesthetized rats at 4 or 24 h after sham or LAD I/R surgery and quickly removed the heart. We sectioned the removed heart into 2 mm-thick slices with a pre-chilled coronal matrix device (HSRA001-1, Zivic Instruments, Pittsburgh, PA, USA). We immersed the coronal sections in a 1% TTC solution in sterile distilled water for 30 min at 37 °C and then fixed them in 4% paraformaldehyde in phosphate-buffered saline for 48 h. First, we scanned each stained section in a flatbed image scanner (PERFECTION V800 PHOTO, EPSON, Nagano, Japan) and measured the infarct volume in each stained section in the image using ImageJ 1.48v software. We determined the infarction area in the anterior and posterior sides of each 2 mm-thick slice in the heart tissue between 0 and 8 mm from the apex of the heart. To measure the infarct volume of each slice, we multiplied the average value of the infarct area on the anterior and posterior sides by thickness (2 mm) (thickness × (top area + bottom area)/2), and we also calculated the total infarct volume as the sum of the infarct volume per slice [10].

### 2.5. Immunohistochemistry Analysis

For immunohistochemistry analysis of the peri-infarction area, first, we performed 2,3,5-TTC staining to confirm the peri-infarction area in the left ventricle. The peri-infarcted region was easily identified between 4 and 6 mm from the apex of the rat heart because it was properly mixed with infarcted and normal tissue in TTC staining. Next, we selected 2 mm-thick slices between 4 and 6 mm from the apex, which were embedded in paraffin after fixing them with 4% paraformaldehyde solution [31]. To acquire segments with 4 µm thickness from a region including the peri-infarct area on New Silane III-coated microslides (Muto Pure Chemical, Tokyo, Japan), we sliced rat heart sections using a microtome (LEICA RM 2335, Wetzlar, Germany). We permeabilized and blocked the sections with 3% H_2_O_2_, citrate buffer, and 5% bovine serum albumin in tris-buffered saline (TBS) for 1 h at room temperature (RT). We incubated the sections in TBS containing Tween 20 and anti-HMGB1 polyclonal primary antibody (1:100, ab18256; Abcam, Cambridge, UK) overnight at 4 °C. We washed the sections three times with TBS for 5 min and incubated them for 1 h at RT using secondary antibodies with fluorescence-conjugated Alexa-fluor 594 (1:100, A11032; Invitrogen, Carlsbad, CA, USA). We also washed the sections three times with TBS and mounted them with ProLongTMDiamond Antifade Mountant containing DAPI (P36962, Invitrogen, Waltham, MA, USA). After immunostaining, we observed and analyzed the peri-ischemic areas of the stained sections with a confocal microscope (LSM 700; Carl Zeiss GmbH, Jena, Germany) [10].

### 2.6. Detection of Intracellular Levels of Reactive Oxygen Species (ROS)

We used cell-permeant 2’,7’-dichlorodihydrofluorescein diacetate (H2DCFDA) stain to detect intracellular ROS levels (Thermo Fisher, Waltham, MA, USA). We deparaffinized the sectioned slide and added 10 μM of H2DCFDA on the slide and incubated it for 15 min at 37 °C in the dark. After H2DCFDA staining, we also observed generation of ROS in the cells using a confocal microscope. In the presence of ROS, H2DCFDA is oxidized to 2’,7’-dichlorofluorescein (DCF), which is detected via green fluorescence, that cannot penetrate the cell membrane. We measured fluorescence intensity using the MetaMorph microscopy automation and image analysis software (Molecular Devices, San Jose, CA, USA) because the fluorescence intensity is proportional to the intracellular ROS level.

### 2.7. Enzyme-Linked Immunosorbent Assay (ELISA) for Detection of Cardiac Troponin T (cTnT) and HMGB1

To obtain rat serum, we punctured the right atrium and obtained blood 4 h after LAD ligation with a 22-gauge needle. We transferred 1 mL of the collected blood into a Z Serum Sep Clot Activator (Greiner Bioone, Kremsmunster, Austria), followed by centrifugation at 3000 rpm for 15 min. We determined the concentrations of cTnT and HMGB1 using a cTnT ELISA kit (MBS2024997, MyBioSource, San Diego, CA, USA) and a HMGB1 ELISA kit (Solarbio, Beijing, China) for rats.

### 2.8. Real-Time Polymerase Chain Reaction (RT-PCR)

To obtain peri-infarcted myocardium tissue, we performed 2,3,5-TTC staining to confirm the peri-infarcted area in the left ventricle [32]. We designed primers for glyceraldehyde-3-phosphate dehydrogenase, IL-1β, IL-6, and TNF-α genes using PrimerQuest (IDT, Skokie, IL, USA). We isolated tissue RNA using a Hybrid-R kit (305-010, GeneAll Biotechnology, Seoul, Korea) and synthesized single-stranded cDNA with 500 ng of total RNA using the PrimeScript 1st strand cDNA synthesis kit (6110A, Takara Bio, Shiga, Japan). We performed quantitative PCR using a 7500 ABI system (Applied Biosystems, Foster City, CA, USA) with SYBR-green (Q5602, Gendepot, Katy, TX, USA) [10].

### 2.9. Terminal Deoxynucleotidyl Transferase (TdT)-Mediated dUTP Nick End Labeling (TUNEL) Assay

To detect apoptotic cells, we conducted the TUNEL assay with the DeadEndTM Fluorometric TUNEL system (Promega, Madison, WI, USA), following the manufacturer’s instructions. We selected and stained one slide from each animal and identified the two peri-ischemic areas of the stained sections using a confocal microscope. We derived the average values of TUNEL-positive cells from two areas in the peri-infarcted area on the stained sections and normalized the numbers of TUNEL-positive cells in the infarcted area using values associated with the sham hearts [10].

### 2.10. Cytokine Array

To construct a cytokine array using a proteome profiler array, we obtained peri-infarcted myocardium tissue after 2,3,5-TTC staining to confirm the peri-infarcted area in the left ventricle 24 h after LAD I/R [32]. We determined the infarct area in the anterior and posterior sides of each 2 mm-thick slice in the heart tissue between 0 and 8 mm from the apex of the heart. We selected peri-infarcted tissue within 3 mm from the end margin of the infarcted tissue in each 2 mm-thick slice between 0 and 8 mm from the apex. We homogenized the peri-infarcted tissue of the rat heart with a homogenizer (Bertin technologies, Montigny, France) in PBS with a protease inhibitor and performed protein quantitation with a BCA kit following the manufacturer’s protocol. We also conducted Proteome Profiler array using the Mouse Cytokine Array Panel A (R&D Systems, Inc., Minneapolis, MN, USA) as per the manufacturer’s instructions, and visualized the blots using the ECL^TM^ Western Blotting Analysis System (GE Healthcare, Chicago, IL, USA) and LAS 4000 mini (Fujifilm, Tokyo, Japan). To analyze the array results, we quantized the blots using HLImage++ (Western Vision software, Salt Lake City, UT, USA) (Appendix A) [10].

### 2.11. Statistical Analysis

We represented the results of all experiments as the mean ± standard deviation (SD) of the mean and conducted statistical analyses using an unpaired *t*-test or one-way analysis of variance (ANOVA), followed by Bonferroni post-hoc tests for multiple comparisons between groups. We considered *p* < 0.05 as a significant difference. We analyzed the post-hoc power using an unpaired *t*-test or ANOVA for each TTC staining, HMGB1 immunohistochemistry, and TUNEL assay.

To identify the explicit difference with five animals/group, after calculating the mean and standard deviation, we assumed that alpha = 0.05. As a result of calculating the post-hoc power, values for all three sets of TTC staining, HMGB1 immunohistochemistry, and TUNNEL assays were 99.99%, exceeding 80%. That was enough to identify the explicit difference.

## 3. Results

### 3.1. Optimal rKL Dose for Myocardial Protection

The mean ratios of the infarcted area at 4 h after myocardial I/R were 20.57 ± 3.23% in the myocardial I/R group without rKL treatment compared to the total area between 0 and 8 mm from the apex of the rat heart. The mean ratios of the infarcted area in the 0.025 µg, 0.05 µg, and 0.075 µg rKL/g of body mass klotho-treated groups after myocardial I/R were 11.3 ± 0.05%, 5.97 ± 3.42%, and 3.71 ± 0.78% after 4 h, respectively (*p* = 0.001). We analyzed the post-hoc power using ANOVA. The infarct volume of the group treated with 0.025 µg rKL significantly differed from that of the myocardial I/R group without rKL treatment (*p* = 0.011). A significant difference was observed in the infarct volume between the 0.025 µg of rKL/g of body mass and 0.05 µg of rKL/g of body mass groups (*p* = 0.027). However, significant difference was not observed in the infarct volume between the 0.05 µg of rKL/g of body mass and 0.075 µg of rKL/g of body mass groups (*p* = 0.730). Thus, we determined that 0.05 µg/g was the optimal dose of rKL for myocardial protection. We subsequently used this rKL dose for myocardial protection (Appendix A).

### 3.2. Administration of rKL Protein Reduces Infarct Volume in Myocardial I/R Injury

In the present study, we assessed infarct volumes by TTC staining after 4 h or 24 h of ischemic reperfusion injury (Figure 1A–E). In the myocardial I/R group without rKL treatment, the mean ratios of the infarct area at 4 h and 24 h after myocardial I/R were 15.72 ± 1.65% and 16.38 ± 6.93% compared to those of the total area between 0 and 8 mm from the rat heart apex, whereas the mean ratio of the infarcted area in the klotho-treated group after myocardial I/R was 5.18 ± 1.40% and 5.22 ± 1.53% after 4 h and 24 h. There was a significant difference between the myocardial I/R group and the klotho-treated group at both 4 h and 24 after myocardial I/R (*p* < 0.001 and *p* = 0.014).

### 3.3. Administration of rKL Protein Reduces Intracellular Levels of Reactive Oxygen Species (ROS)

We assessed intracellular ROS levels using H2DCFDA staining after 4 h. After myocardial I/R, intracellular ROS levels, which were represented by fluorescence intensities, were significantly reduced in the klotho-treated myocardial I/R injured group (8.98 ± 6.84) compared to the myocardial I/R injured group (40.70 ± 8.59) (*p* < 0.001) (Figure 2A,B).

### 3.4. Administration of rKL Protein Attenuates the Extracellular Release of HMGB1 from Peri-infarct Tissue after Myocardial I/R Injury and Effects of the Administration of rKL Protein on HMGB1 Levels in the Plasma

When ischemic damage to the myocardium is induced by LAD ligation in the heart, HMGB1 is released from the nucleus of myocardial cells [33,34]. We identified that HMGB1 immunoreactivity was significantly diminished in the myocardium after LAD ligation in rat hearts. To investigate whether the rKL protein significantly diminishes the extracellular release of HMGB1 following I/R injury, we compared immunoreactivity of HMGB1 between the myocardial I/R injury and klotho-treated groups after myocardial I/R injury. We found that 18.25 ± 15.04% of 4,6-diamidino-2-phenylindole (DAPI)-positive cells in the peri-ischemic myocardium of rats with LAD ligation were also positive for the expression of HMGB1. However, we also found that klotho-treated rats demonstrated restoration of the number of HMGB1-positive cells in the infarct tissue after I/R injury. The percentages of HMGB1-positive cells were 79.45 ± 5.68% after administration of rKL. Significant increases were observed in the proportion of HMGB1-positive cells in the klotho-treated myocardial I/R group compared to that in the myocardial I/R group (*p* < 0.001). This suggests that the administration of rKL protein significantly reduced HMGB1 extracellular release in myocardial damage after I/R injury (Figure 3A,B). In addition, to measure HMGB1 levels in serum samples, we conducted ELISA with blood samples obtained at 4 h after the onset of ischemia. As expected, myocardial I/R injury increased the level of circulating HMGB1 (104.06 ± 32.18 pg/mL). However, klotho treatment significantly reduced the increase in the levels of circulating HMGB1 ((34.34 ± 10.98 pg/mL), *p* < 0.001) (Figure 3C).

### 3.5. Effects of the Administration of rKL Protein on cTnT

To identify the myocardial protective effects of klotho protein on cTnT levels reflecting myocardial damage, we determined cTnT levels in the plasma. The cTnT levels were higher in the myocardial I/R injury group in comparison with those in the sham-operated group (3.78 ± 0.99 and 0.37 ± 0.19 ng/mL, respectively; *p* < 0.001). However, rats treated with rKL protein showed lower cTnT levels than those in the myocardial I/R injured group (1.33 ± 0.09 and 3.78 ± 0.99 ng/mL, respectively; *p* < 0.001). No significant difference was observed in plasma cTnT between sham-operated and klotho-treated groups (*p* = 0.11), indicating that klotho treatment reduced myocardial damage (Figure 3D).

### 3.6. Administration of rKL Protein Inhibited Expression of Pro-Inflammatory Cytokines from Peri-Infarct Regions

We assessed the cardiac mRNA expression of three major inflammatory cytokines (i.e., TNF-α, IL-1β, and IL-6) in the peri-infarcted myocardium 4 h after LAD ligation in the rat heart, using quantitative RT-PCR. The expression levels of TNF-α (18.71 ± 12.28, *p* = 0.031), IL-1β (73.32 ± 25.21, *p* < 0.001), and IL-6 (3986.57 ± 3176.19, *p* = 0.045) were significantly increased in rats with myocardial I/R injury. Klotho-treated rats with myocardial I/R injury demonstrated a decrease in the expression of these cytokines in the peri-infarcted area after myocardial I/R injury (TNF-α (4.60 ± 0.50, *p* = 0.031), IL-1β (23.39 ± 3.44, *p* < 0.001), IL-6 (550.19 ± 193.98, *p* = 0.045)) in comparison with rats with myocardial I/R injury (Figure 4A–C). Thus, the administration of recombinant klotho protein attenuated the aggravation of myocardial damage via suppression of the production of inflammatory cytokines in the peri-infarcted myocardium after I/R injury.

### 3.7. rKL Protein Attenuates Apoptosis in the Myocardium After Myocardial I/R Injury (TUNEL Assay)

The number of TUNEL-positive apoptotic cells, represented by light green dots under the confocal microscope, was remarkably increased in the myocardial I/R injury group (93.53 ± 21.86) compared to the klotho-treated myocardial I/R injury group (34.13 ± 9.13, *p* < 0.001). These results also imply that the application of recombinant klotho protein demonstrated significant myocardial protection effects by attenuating apoptosis induced by myocardial I/R injury (Figure 5A,B).

### 3.8. Administration of rKL Protein Modulated Cytokine Production in the Myocardial I/R Injury

To identify and compare the expressions of specific cytokines according to rKL protein treatment, we performed cytokine array analysis with the whole heart tissue at 24 h after myocardial I/R injury (Figure 6A). We identified that the expressions of several cytokines increased in the heart following myocardial I/R injury. In comparison with the myocardial I/R injury group not treated with rKL, administration of rKL protein significantly reduced the expression levels of several cytokines, such as cytokine-induced neutrophil chemoattractant 1 (CINC-1), intracellular adhesion molecule-1 (sICAM-1/CD54), C-X-C motif chemokine 5 (CXCL5; LIX), and L-selectin (Figure 6B). Thus, we found that rKL protein regulated the levels of inflammation-related proteins. Appendix A showed the statistical results of the present study (Appendix A).

## 4. Discussion

In our study, we evaluated the therapeutic effects of rKL using an established rat model, similar to those observed with recanalization after AMI. First, administration of rKL protein significantly reduced infarct volume in rats with myocardial I/R injury. After myocardial I/R injury, the increase in the levels of cardiac troponin, which is a marker of heart muscle damage, was significantly inhibited by rKL protein.

Although inflammation is fundamental in wound healing after AMI, cardiomyocytes can be damaged significantly with a potent inflammatory response induced against I/R injury [8,10]. Cell death through a combination of necrosis and apoptosis begins early after the cessation of blood flow (within 30 min to 1 h) [8,10]. The onset of ischemia initiates inflammation, which leads to devastating myocardial damage. Reperfusion also sustains inflammation over several hours [8,10]. During cell injury in the absence of infectious threats, such as ischemia, radiation, trauma, and autoimmune disease, HMGB1 is released as an early mediator that balances the release of several cytokines, such as TNF, and tissue damage via molecular mechanisms involving signalling via binding of HMGB1 and TLR4 [15]. Therefore, it is important to mitigate the early effects of HMGB1 to attenuate I/R injury of the myocardium [10]. This study was the first to suggest the possibility of a functional and mechanistic link between the administration of rKL and HMGB1 in a clinically relevant acute myocardial I/R animal model. After acute myocardial I/R injury, administration of rKL protein could attenuate sterile inflammation by inhibiting extracellular HMGB1 release in peri-infarcted regions and subsequently reducing myocardial damage, resulting in effective protection of the myocardium after I/R injury. In addition, rKL protein suppressed the production of inflammatory cytokines associated with signalling, involving the binding of HMGB1 and TLR4 in the peri-infarct area after myocardial I/R injury. The rKL protein attenuated extracellular HMGB1 release and reduced the mRNA expression of pro-inflammatory cytokines, such as TNF-α, IL-1β, and IL-6, in I/R-injured rat hearts, which is similar to the effects of the HMGB1 inhibitor glycyrrhizin [10]. Interestingly, rKL treatment after myocardial I/R injury demonstrated protective effects on the myocardium, which were similar to those of glycyrrhizin treatment before I/R injury in the rats (unpublished data) [35,36,37]. Our finding suggests that rKL helps to prevent this devastating propagation of damage by inhibiting the extracellular release of HMGB1 after myocardial I/R injury, and inhibition of HMGB1 by rKL accounts for its suppression of peri-infarct inflammation. Further, glycyrrhizin showed a myocardial protective effect in pretreatment; however, it did not show the same effect after I/R injury [35,36,37]. Unlike glycyrrhizin, rKL administration demonstrated effective myocardial protection after I/R injury. This finding showed the potential of klotho as a therapeutic agent. The findings of this study support the new hypothesis that administration of rKL acts to inhibit HMGB1 in acute myocardial I/R injury. rKL subsequently attenuates sterile inflammation after I/R injury. This is the first study to suggest that exogenous administration of rKL exerts myocardial protection effects after I/R injury and provides novel mechanistic insights into rKL that can provide a theoretical basis for the clinical application of new adjunctive modalities for the critical care of AMI.

CXCL5 is a powerful chemo-attractant of neutrophils and pro-angiogenic factors involved in the process of the innate immune response. CINC-1 plays a critical role by attracting neutrophils to the site of inflammation in neutrophil-mediated inflammatory diseases [38]. ROS are indispensable messengers, and NF-κB is a ubiquitous and redox-sensitive transcription factor that plays a critical role in CINC-1 production [38]. Cytokine-induced neutrophil chemoattractant 1 signalling pathway for CINC-1 production is significantly associated with NF-κB-mediated transcription and ROS generation in the mitochondria [38]. HMGB1-TLR4 signalling upregulates the expression of cytokines and other inflammatory mediators by activating MyD88-dependent nuclear translocation of NF-κB. Our findings demonstrated that rKL suppressed the extracellular release of HMGB1 and ROS production after I/R injury. In consequence, this may lead to attenuation of the activation of CINC-1 signalling. L-selectin regulates leukocyte adhesion and recruitment to lymph nodes in the periphery and sites of acute and chronic inflammation [39]. IL-1 and TNF can induce the expression of ICAM-1, which is expressed by vascular endothelium, macrophages, and lymphocytes [40]. Leukocytes adhere to endothelial cells via ICAM-1/LFA-1, thereby penetrating the tissues [40]. CXCL5 is also a powerful chemoattractant of neutrophils and pro-angiogenic factors involved in the process of the innate immune response [41]. The production of CXCL5 is similar to that of ICAM-1, which is mediated by the stimulation of cells with IL-1 or TNF that act as inflammatory cytokines [41]. Our study demonstrated that rKL protein significantly reduced the production of the protein levels of several cytokines, such as CXCL5: LIX, CINC-1, intracellular adhesion molecule-1 (sICAM-1/CD54), and L-selectin.

As klotho prevents oxidative stress, senescence, and apoptosis, decreased klotho levels have been significantly associated with an increase in oxidative stress. Oh et al. suggested that klotho plays a role in preventing apoptosis progression via antioxidative effects in the contrast-associated acute kidney injury (CA-AKI) model [22]. Moreover, we also showed that exogenous supplementation of rKL may attenuate progressive apoptosis via the antioxidative effect by reducing intracellular ROS levels in the early stage of I/R injury. In this study, the activity of ROS was significantly attenuated after 4 h of rKL supplementation. Oh et al. demonstrated that rKL supplementation significantly decreased ROS activity after 16 h, but not at 24 h, in the CA-AKI model [22]. The antioxidative effect of rKL supplementation may be limited over time after I/R injury. rKL may be supplemented in the early stages after I/R injury to maximize the antioxidative effect. We also found that the administration of rKL exerted significant myocardial protective effects against cellular apoptosis, compared to those observed in rats with myocardial I/R injury not treated with rKL. We identified that rKL intuitively attenuated cell death in the myocardium, with significant improvements in aggravation of the infarcted area that lasted up to 24 h.

Similar to the present study, our previous study showed results using a clinically relevant rat model of AMI that targeted temperature management inhibition of infarct volume expansion by reducing inflammatory cytokines through the blockage of HMGB1 release in the peri-infarcted region after myocardial I/R injury [10]. The hypothermic benefits throughout the body can significantly be minimized by adverse effects, such as hypotension, lethal arrhythmia, infection, and shivering [41]. Since systemic hypothermia can affect all organ systems in the human body, the use of TTM should be based on the balance between its potential benefits and possible harmful effects [42]. Until now, longer and deeper hypothermic conditions were required to maximize the benefits of TTM, which required sedation, anti-shivering agents, advanced airway management, and mechanical ventilation [43,44], which may be harmful in practice [43]. Although this is a preclinical study, administration of rKL showed therapeutic effects similar to those of TTM and may be an effective alternative for ameliorating myocardial damages in patients with AMI who are awake and exhibit spontaneous breathing.

This study has several limitations. Although protective effects similar to those observed with glycyrrhizin treatment were reported, explicit evidence demonstrating a direct association between rKL and HMGB1 inhibition is lacking. In future research, to explore the specific mechanism of rKL, it would be necessary to elucidate a mechanism by comparing the effects of rKL with those of direct antibodies or substances with a known mechanism. In addition, it is very difficult to demonstrate the pharmacokinetics of exogenously administered rKL based on a clinically relevant acute myocardial I/R animal model. We need to demonstrate whether rKL is integrated into the cells or promotes the expression of KL in cells. The final goal of this study was to find a drug that is helpful in the treatment and prognosis of patients with myocardial infarction in actual clinical practice. Therefore, klotho injection was administered with a focus on reperfusion in this study. In future experiments, studies on the time and administration route that can optimize the effects for the treatment of reperfusion injury should be conducted.

Supplementation of rKL can be a viable and adjunctive therapeutic modality for patients with I/R injury, including those with out-of-hospital cardiac arrest, hemorrhagic shock, AMI, and ischemic stroke. In addition, the positive effect of klotho observed in various organs can be expected to play an important role in the treatment of AMI in the future. However, in-depth studies are needed to validate the clinical benefits of rKL supplementation and the mechanisms for myocardial protection of rKL against sterile inflammation in patients with AMI.

## 5. Conclusions

Our study suggested that the administration of rKL exerted myocardial protective effects by attenuation of the extracellular release of HMGB1, inflammation, apoptosis, production of intracellular ROS, and cell death after I/R injury. This study provides insights into the mechanism of rKL that can provide the theoretical basis for clinical applications of new adjunctive modalities for the critical care of AMI.

## Figures and Tables

**Figure 1 biomedicines-10-00894-f001:**
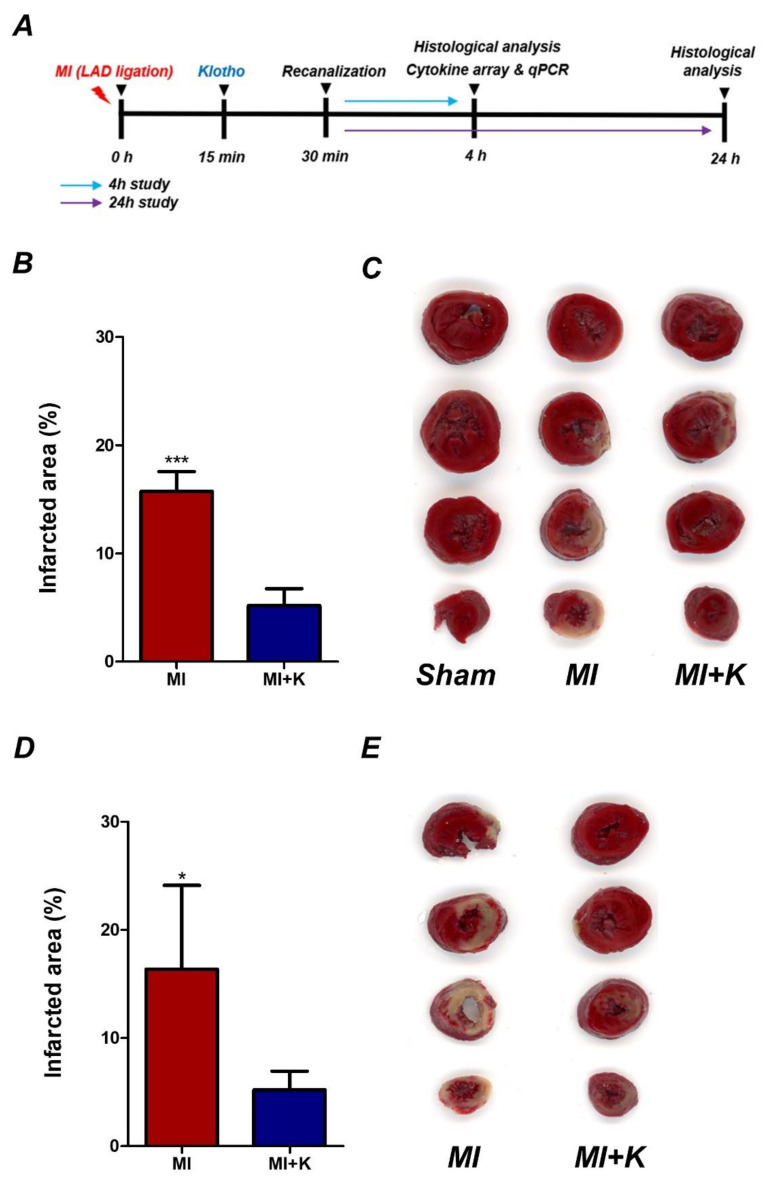
Recombinant klotho (rKL) reduces infarct volume after myocardial I/R injury. (**A**) Experimental schedule; (**B**) Volume of myocardial infarct area stained with of 2,3,5-triphenyltetrazolium chloride (TTC) at 4 h, *** *p* < 0.001, comparison of myocardial I/R with rKL, one-way analysis of variance (ANOVA) followed by Bonferroni post hoc test. (**C**) Representative image of 2,3,5-triphenyltetrazolium chloride (TTC) staining at 4 h; (**D**) Volume of myocardial infarct area stained with TTC observed at 24 h, * *p* < 0.05, comparison of myocardial I/R with rKL, unpaired *t*-test. (**E**) Representative image of 2,3,5-triphenyltetrazolium chloride (TTC) staining at 24 h (the number of animals: *n* = 5, respectively).

**Figure 2 biomedicines-10-00894-f002:**
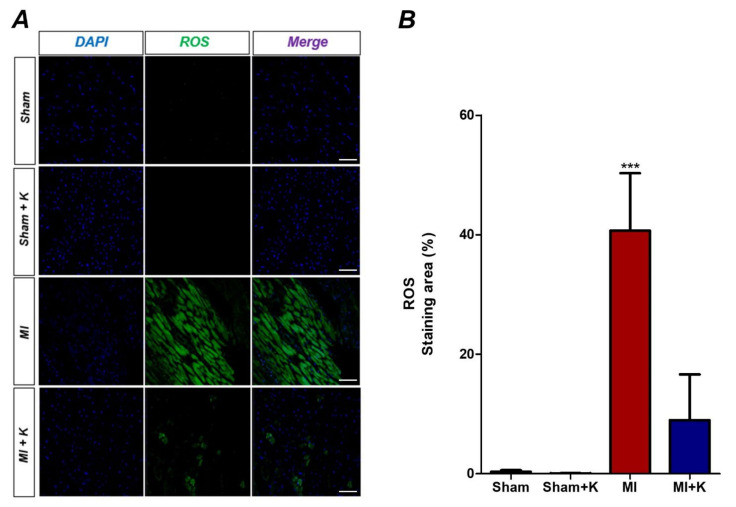
Intracellular reactive oxygen species (ROS) level. (**A**) Representative intracellular reactive oxygen species (ROS) results observed following administration of klotho after myocardial ischemic/reperfusion (I/R) injury; (**B**) ROS staining area, *** *p* < 0.001, comparison of myocardial I/R with and without administration of recombinant klotho (rKL) by one-way analysis of variance (ANOVA) followed by Bonferroni post hoc test (number of animals: *n* = 5). Scale bar, 50 µm.

**Figure 3 biomedicines-10-00894-f003:**
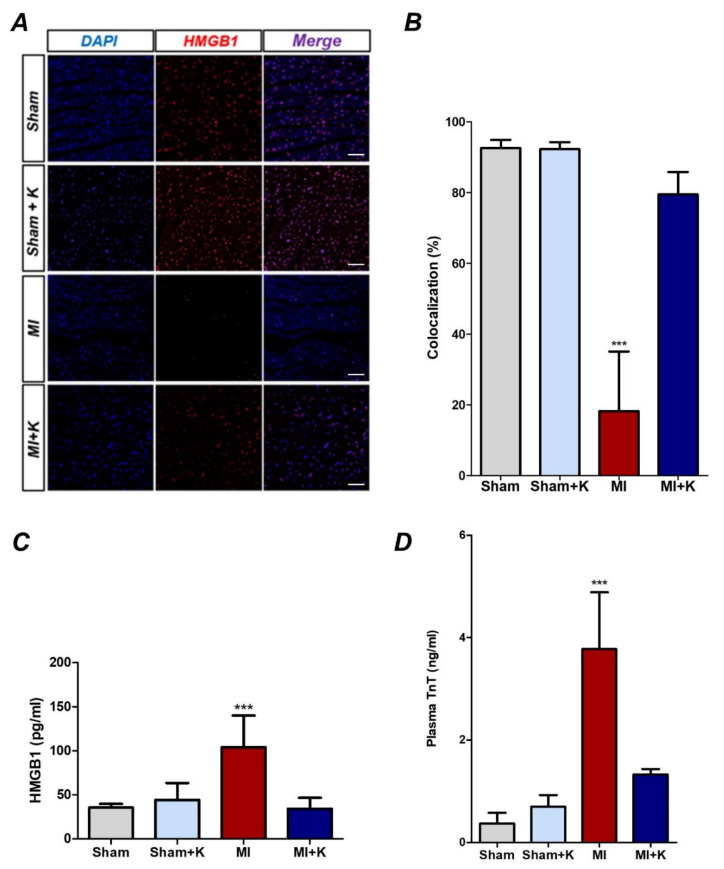
Recombinant klotho (rKL) suppresses the extracellular release of high mobility group box-1 (HMGB1) after myocardial ischemic/reperfusion (I/R) injury. (**A**) Representative immunohistochemistry results for rKL treatment after myocardial I/R injury; (**B**) Immunohistochemistry results, *** *p* < 0.001, comparing myocardial I/R with rKL using one-way analysis of variance (ANOVA) followed by Bonferroni post hoc test (the number of animals: *n* = 5, respectively). (**C**) Levels of circulating of HMGB1 in plasma, *** *p* < 0.001, comparison of myocardial ischemic/reperfusion (I/R) injury with and without recombinant klotho (rKL) administration by one-way analysis of variance (ANOVA) followed by Bonferroni post hoc test (the number of animals: *n* = 5, respectively). (**D**) Levels of cardiac troponin T (cTnT) in plasma, *** *p* < 0.001, comparison of myocardial ischemic/reperfusion (I/R) injury with and without recombinant klotho (rKL) administration by one-way analysis of variance (ANOVA) followed by Bonferroni post hoc test (the number of animals: *n* = 5, respectively). Scale bar, 50 µm.

**Figure 4 biomedicines-10-00894-f004:**
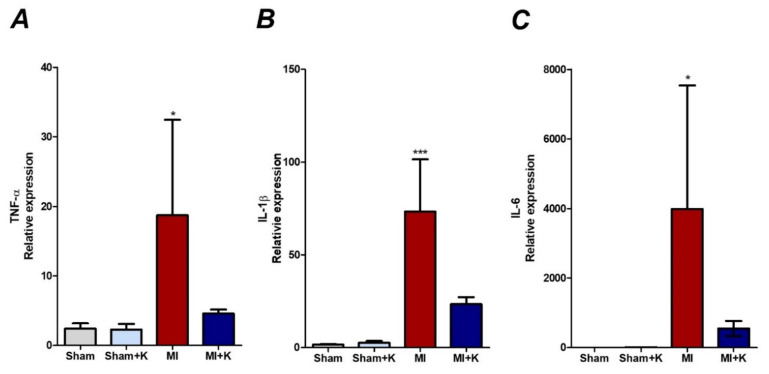
Cardiac mRNA expression of three major inflammatory cytokines in the peri-infarcted regions. (**A**) Quantification of the expression of tumor necrosis factor-α (TNF-α) by RT-PCR, * *p* < 0.05, comparison of myocardial ischemic/reperfusion (I/R) injury treated with klotho by one-way analysis of variance (ANOVA) followed by Bonferroni post hoc test. (**B**) Quantification of interleukin-1β (IL-1β) expression by RT-PCR, *** *p* < 0.001. Comparison of myocardial I/R injury treated with recombinant klotho (rKL) by ANOVA followed by Bonferroni post hoc test. (**C**) Quantification of IL-6 expression by RT-PCR, * *p* < 0.05. Comparison of myocardial I/R treated with klotho by ANOVA followed by Bonferroni post hoc test (the number of animals: *n* = 5, respectively).

**Figure 5 biomedicines-10-00894-f005:**
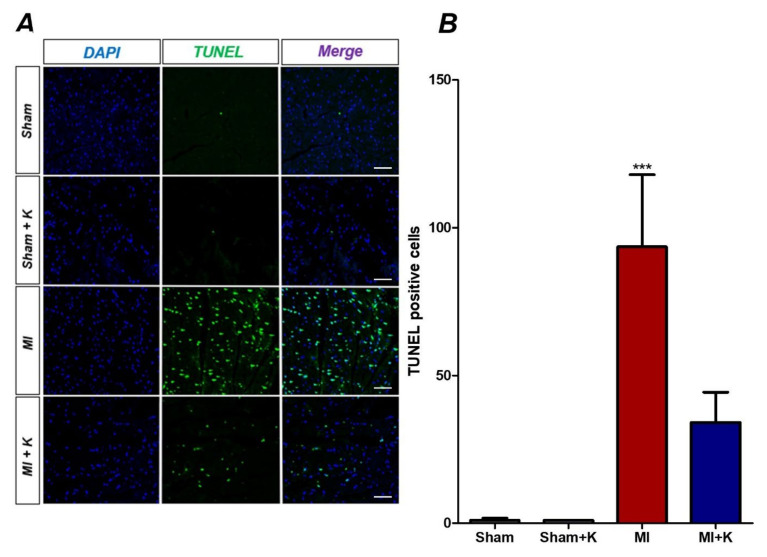
Terminal deoxynucleotidyl transferase (TdT)-mediated dUTP nick end labeling (TUNEL) assay. (**A**) Representative TUNEL assay results for recombinant klotho (rKL) administration after myocardial ischemic/reperfusion (I/R) injury; (**B**) TUNEL assay results, *** *p* < 0.001, comparison of myocardial I/R treated with rKL by one-way analysis of variance (ANOVA) followed by Bonferroni post hoc test. Scale bar, 50 µm.

**Figure 6 biomedicines-10-00894-f006:**
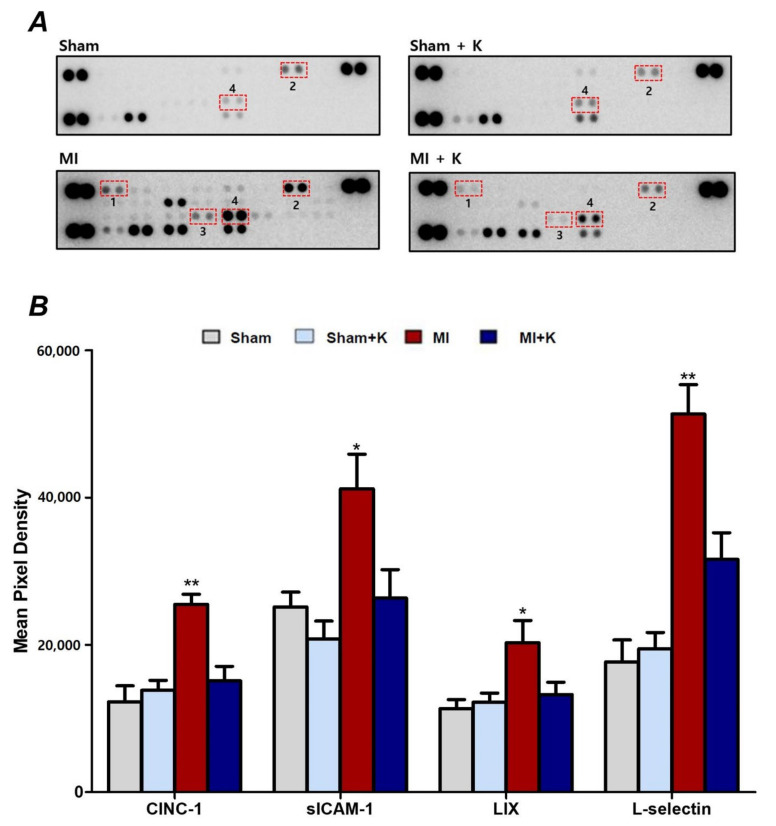
Expression levels of several cytokines. (**A**) Representative chemiluminescence images of the proteome profiler array after administration of klotho following myocardial I/R injury. The dotted line box indicates the selected candidate cytokines. (**B**) Expression levels of several cytokines, * *p* < 0.05, ** *p* < 0.01, comparison of myocardial ischemic/reperfusion (I/R) injury with and without recombinant klotho (rKL) administration by one-way analysis of variance (ANOVA) followed by Bonferroni post hoc test (the number of animals: *n* = 5, respectively).

## Data Availability

Not applicable.

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
