# Peer review of "Recombinant Klotho Protein Ameliorates Myocardial Ischemia/Reperfusion Injury by Attenuating Sterile Inflammation"

_biomedicines, 2022, doi:10.3390/biomedicines10040894_

Round 1

Reviewer 1 Report

This study examined the effects of rKL on myocardial infarct size in the rat model of ischemia-reperfusion. Decreased myocardial necrosis was found mediated by attenuated inflammation, apoptosis and production of intracellular ROS. The topic is of high clinical interest and this work adds to current understanding. The experiments appear to have been meticulously performed and the results are presented clearly. Overall, the manuscript is well written. A few points need to be further addressed.

1) The main drawback of the study is the small number of experiments (it seems that n=5 in most comparisons) and the consequent possibility of alpha error. To partly overcome this limitation, please provide the statistical results in more detail, ie mean values, standard deviation and p values for every comparison. Likewise, please provide such data for the dose-finding study (lines 135-140).

2) The statement ‘nevertheless, in-hospital mortality has not been reduced significantly in patients with STEMI undergoing primary PCI’ (line 42) is not accurate. Moreover, the major goal of myocardial protective therapies post-MI is not short-term mortality but the attenuation of left ventricular remodeling and, hence, the progression to chronic heart failure. I believe that this part of the introduction should be revised.

3) In the discussion, I think that the comparison between the results of the present study and those of the previous study by the same group (Ref #10) should be discussed in more detail, with relevant implications on pharmacologic actions.

4) Line 367: ‘we established a rat model to evaluate the therapeutic effects of rKL’. I believe that the authors mean ‘we evaluated the therapeutic effects of rKL using an established rat model’.

Author Response

Please see the attachment (marked copy and response to reviewer's comments).

We appreciate your comments and useful suggestions.

Reviewer 2 Report

Introduction

Lines 42-44: Given the density of abbreviations used (> 9) on the mentioned pages and throughout the paper, an Abbreviation list would be desired.

          The notion of sterile inflammation is used copiously in the Introduction. However, it would be interesting for the authors to separate it from inflammatory reaction occurring during ischemia-reperfusion.

Lines 94-98:  The expression “We aimed” is used repeatedly, and the whole paragraph is monotone, not to say repetitive. Please correct this.

 Materials and Methods

Lines 136-140: Contain results and should be placed under Results instead of Material and Methods.

Lines 228-235: Was the protein determination done on the whole homogenate? Or was the homogenate clarified before? A better protocol is desired.

Line 240: Who is SHE who analyzed the post-hoc power? Is this real? It must be clarified.

          How many rats were used in each of: 2.4; 2.5; 2.6; 2.7; 2.8; 2.9 and 2.10? Their # should match the one from the groups from 2.3.

Discussions

Lines 377-378: What are “ other forms of sterile cell injury”? A new concept, a new field of investigation, the same as the sterile inflammation from Introduction? Please clarify and remain consistent throughout the paper.

 As much as the authors used glycyrrhizin in their previous published work, why are they bringing the “glycyrrhizin” into the discussion (Lines 389-394)? Did they do any experiments to compare their treatment with the use of glycyrrhizin? Unfortunately, this work has no data to back up such a claim.

Author Response

(The authors gave the same response as above.)
